# From Fashion Brand to Artwork: Divergent Thinking, Copyright Law, and Branding

**Marlena Jankowska [1,\*] and Berenika Sorokowska [2,\*]**

1   Faculty of Law and Administration, University of Silesia in Katowice, PL-40-007 Katowice, Poland
2   Faculty of Law and Administration, The University of Warsaw, PL-00-927 Warszawa, Poland
\*   Correspondence: marlena.jankowska@us.edu.pl (M.J.); berenika.sorokowska@gmail.com (B.S.)

**Abstract:** The purpose of this study is to explore the interaction between copyright, branding, marketing, and heritage protection with regard to a fashion brand. The authors use analytical-critical and legal-dogmatic methods, supplemented with desk research, a case study approach, and a review of the marketing literature. This paper argues that the top-tier fashion brands use the concept of artification in order to build their brands, mesmerize clientele, and increase revenues. Although design and reference to the arts play a major role in the luxurious and premium end of the fashion business, this analysis proves that the top players do not necessarily observe the appropriate laws in these areas. The reader will see examples of the flouting of basic legal constraints by big players, e.g., copyrights or property rights, including the monetisation of the creativity of others with the expectation of no legal challenge. Offenders capitalise on the likelihood that a legal suit is too demanding for smaller players, such as foundations or museums.

**Keywords:** fashion; art; brand; copyright law; cultural heritage; European Union law; art appropriation

## 1. Introduction

Divergent thinking (Baer 2014; Acar and Runco 2019; Hocevar 1980; Lewis and Lovatt 2013; Nusbaum et al. 2014; McCrae 1987) is a way of thinking that involves coming up with creative and unconventional solutions to problems by exploring a wide range of possible solutions. It is the ability to think outside the box, to deviate from traditional and linear ways of thinking, and to generate a wide range of potential solutions to a problem.

Divergent thinking is frequently linked to creativity, innovation, and problem-solving abilities. Convergent thinking, on the other hand, involves narrowing down ideas to find the single best solution to a problem.

Brainstorming, mind mapping, and free association are some techniques that can help promote divergent thinking. These techniques encourage people to generate as many ideas as they can without judging or criticising each other, and to build on each other's ideas. In a way, divergent thinking serves to democratise the heritage of art and fashion. Both fields are, as it were, condemned to each other, inseparable and interdependent, also in the copyright regime. Divergent thinking in both fields is inevitable, all the more so as the trend of building a strong brand position based on art is also increasingly recognised in the marketing literature.

Branding, cultural heritage, and copyright are all related to the protection and promotion of intellectual property, but in slightly different ways (World Trade Organisation 2017; Caponigri 2021). Branding refers to the use of a name, term, design, symbol, or other feature to identify and distinguish a product or service from those of other companies. A strong brand can create customer loyalty, increase market share, and add value to a business. In this sense, branding can be seen as a way of protecting a company's reputation and intellectual property. Branding will be examined only from the economic sciences' point of view, leaving discussions related to trademarks outside of the scope of this paper.

Cultural heritage refers to the traditions, customs, beliefs, and artefacts that are passed down from generation to generation within a particular society or group. Cultural heritage is often considered to be a source of pride and identity for a community, and can include everything from language and music to architecture and art. Protecting cultural heritage involves preserving and promoting these traditions for future generations. On the other hand, copyright is a legal concept that grants the creator of an original work (such as a book, song, or film) exclusive rights to control how the work is used and distributed. Copyright law is intended to protect the moral and economic interests of creators by preventing others from using their work without permission or compensation.

The intersections can be delineated in the following ways:

1. **Branding vs. Cultural Heritage** (Hakala et al. 2011; Amer 2018; Nobre and Sousa 2022): A brand is more than just a logo or a name. It represents a company's values, culture, and history. Companies often use cultural heritage as part of their brand identity, drawing on the unique characteristics of cultural origins to create a distinct brand image. For example, a company that is based in a particular region may use local cultural symbols, language, or traditions to create a distinct and memorable brand.[1]

2. **Cultural Heritage vs. Copyright** (Corbett and Boddington 2011): Cultural heritage is often subject to intellectual property law. Copyright protects original works of authorship, including literary, musical, and artistic works that did not fall into the public domain (author's life plus 70 years). This means that cultural artefacts, such as traditional songs, dances, or crafts, cannot be copyrighted by their creators or owners. Ownership rights help to protect the cultural heritage (Corbett 2012) of a community or objects belonging to individuals, preventing unauthorised use or exploitation of cultural works.

3. **Branding and Copyright** (Matenaer 2023; Kiser 2016; Bomsel 2013): Branding often involves the creation and use of copyrighted works, such as logos, slogans, or marketing materials. Companies must ensure that their branding activities do not infringe on the copyrights or trademarks of others. This means that they may need to obtain permission or licenses to use copyrighted works or create their own original works that are not subject to copyright protection. Trademark law stays outside of the scope of this paper; however, it should be noted that copyright law offers protection for unregistered trademarks as well.

All three of these concepts can intersect in various ways. For example, a company may use cultural heritage elements in its branding to create a sense of authenticity or cultural relevance. Copyright law and laws of cultural heritage can also be used to protect the use of cultural heritage materials, such as designs, from being used without permission or compensation. Finally, promoting and protecting cultural heritage can be an important part of a company's branding strategy, especially if it is seen as being socially responsible or supporting a particular community. Companies must navigate copyright laws carefully when creating and using branded materials that may be subject to copyright protection.

Branding, cultural heritage, and copyright are all highly relevant in the fashion industry, and together can play a significant role in how fashion is designed, marketed, and protected. Fashion designers often draw inspiration from traditional clothing and textiles from different cultures around the world. For example, African wax prints, Japanese kimono fabrics, and Indian embroidery techniques have all been incorporated into contemporary fashion designs. As acknowledged by Palandri "Especially in the *haute couture*, but also in the *prêt-à-porter*, stylists and artists have constantly been a mutual source of inspiration, influencing and stimulating each other's innovation and creativity. From the historical Mondrian day dress by Yves Saint Laurent and Souper dress by Andy Warhol in the 1960s,

---

[1] As noted "it has been referred that a branding assists to upgrade the quality of the heritage destination; to form socially a linkage between the destination stakeholders; and to create the additional distribution channels. Thus, a branding adds a new symbolic added-value associating the commercial purpose with the real function of a cultural heritage".

through the incursions of artists Jackson Pollock, Dan Flavin, Sol Lewitt, and Robert Morris into the fashion magazines, the relationship between fashion and art flourished in the early 90s with designers such as Rei Kawakubo of Comme des Garçons, Hussein Chalayan, Martin Margiela, and Alexander McQueen. It is, then, only in the last fifteen years that thriving collaborations between fashion brands and artists have exponentially increased" (Palandri 2020).

Companies can generate a sense of nostalgia and authenticity in their branding by using cultural heritage.

The common ground between these concepts in the fashion industry is the importance of creating and protecting valuable intellectual property. Fashion brands that are successful at branding—drawing inspiration from cultural heritage and protecting their designs through copyright—can gain a competitive advantage and establish themselves as leaders in the industry. At the same time, it is important to balance commercial interests with cultural sensitivity and respect for the origins of traditional designs (Derclaye 2010).

These three concepts intertwined together can trigger many legal issues that need interpretation. However, referencing European Union law or national regulations, they are not easily solved (Ambrosino 2016; Höpne and Schäfer 2012). In the context of European and national copyright law, heterogeneity can refer to the various legal frameworks and rules that exist across EU member states for copyright protection and enforcement. Many countries, including the EU, have some form of copyright protection known as moral rights. These rights confer non-economic rights to authors, such as the right to be identified as the author, and the right to prevent derogatory treatment of their work.

The Berne Convention for the Protection of Literary and Artistic Works,[2] an international treaty that establishes the fundamental principles of copyright protection, recognises moral rights in the EU. The EU Copyright Directive[3] contains provisions on moral rights as well.

However, how moral rights are implemented and enforced varies across EU member states. Some member states, for example, may recognise a broader range of moral rights than others, or may have different procedures in place to enforce those rights.

Even though the EU Copyright Directive governs copyright law in the EU, laying out the fundamental principles of copyright protection across member states, each member state, however, is free to implement the directive in its own way, resulting in a patchwork of national laws that can differ significantly. As the Advocate General, M. Szpunar, emphasised in his opinion in the Cofemel case, "it is true that important aspects of copyright fall outside the scope of Directive 2001/29: moral rights, the collective management of rights, the defence of those rights (except the general provisions of Article 8)" (Szpunar 2019). Since a work of authorship is a bundle of two kinds of copyrights, moral and economic, it should be stressed that these strongly correlate with each other. In other words, when an infringement upon a work occurs, typically, it can be gauged from more than just one perspective and will be an infringement of more than one kind. This interrelation is acknowledged in the legal practice regardless of whether the case is ruled under a monistic or dualistic system. Since member states can tailor the laws to their own legal traditions and cultural needs, the dualistic system allows for some flexibility (Hugenholtz and Senftleben 2011) in the implementation of copyright law. It may, however, result in differences in how copyright law is enforced and interpreted across member states.

Overall, while copyright law provides a framework for protecting creative works, the EU legal system's heterogeneity means that there may be some differences in how copyright is enforced and applied in different member states (Rosati 2021). The attempt made to confront this issue in the fashion industry will become a pretext for other, further considerations.

---

2   Berne Convention for the Protection of Literary and Artistic Works, 9 September 1886, as revised at Paris on 24 July 1971 and amended in 1979; (Díaz 2010).

3   The EU copyright law consists of 13 directives and 2 regulations, harmonising the essential rights of authors, performers, producers and broadcasters; (European Commission 2023).

## 2. "I Am Not Commercial . . . I Am an Artist" ([New York Times 1913](#)): Fashion Brand Identity

The fashion industry, especially at the upper end of the scale, is very prone to any influence from the arts, and these happen at many levels. In many cases, the creative process does not start with the concept for the good or service itself, but rather with the brand identity, the creation of a distinct brand personality or brand concept that will allow the consumer to recognise the brand and its products, to not only feel comfortable with his purchase, but also to yearn for it and to make the product a part of his own self-image. Customers provide brands with mental space in their heads (brand share) based on which special feelings they generate, such as love, affection, addiction, or even hate. The brand effect is so potent that it effortlessly permeates any fashion item, regardless of how creative or market-specific it is[4] ([Nässel and Persson 2011](#); [Geiger-Oneto et al. 2013](#)). Miuccia Prada and Patrizio Bertelli coined the term "fashionless fashion" ([Masè and Silchenko 2017](#)), which denotes the phenomenon of a strong brand that offers not very sophisticated goods at exorbitant prices, and letting the clientele love the product more because of the love for the brand itself overlooking any deficiencies in the product's features.

This triggers a lingering question as to the nature of this powerful phenomenon called 'brand'. The law can offer no ready legal definition and instead advances rather inconclusive ideas of what it is, mostly pointing at reputation, good name, and goodwill ([Jankowska and Pawełczyk 2023](#)). However, there is a great deal of discourse in the marketing studies with regard to branding and 'brand' alone. As for branding, Padela, Wooliscroft, and Ganglmair-Wooliscroft offer four perspectives, which serve as an overarching umbrella for a variety of 'brand' concepts. They assert that branding may be product- and firm-centric, consumer-centric, relational, or sociocultural ([Padela et al. 2023](#); cf. [Brodie et al. 2017](#)). This umbrella covers multiple approaches and attempts to decode what a brand is: a **corporate brand** ([Knox and Bickerton 2013](#)), **brand equity and value** ([Keller 2013](#)), and **brand identity** ([Aaker 1996](#)), symbolic transaction ([Berger and Heath 2007](#); [Escalas and Bettman 2005](#)), **brand awareness**, **brand image** ([Biel 1997](#); [Keller 2003](#)), **brand personality** ([Aaker 1996](#)), and **customer-based brand equity** ([Keller 1993](#)), **brand engagement** ([France et al. 2016](#)), **cultural hotspot** ([Diamond et al. 2009](#)), **brand heritage** ([Rose et al. 2016](#)), **brand culture** ([Schroeder and Salzer-Mörling 2006](#)), and **brand iconicity** ([Holt 2004](#)). Sociocultural branding is especially of essence here, as, in this perspective, a brand is viewed as a cultural resource, including all of its myths and associations. A similar idea was developed by Bergvall, who emphasises the brand's halo effect and cultural component, which turns the brand into a cultural artefact ([Bergvall 2006](#)). Conclusively, brands adapt over time to reflect changes in the sociocultural, community, and ideological values.

## 3. 'Artification' of Fashion

Prominent fashion brands profit from their acclaimed position by "ratifying" their goods ([Masè and Cedrola 2017](#)). An approach based on the arts called "artification" raises people's perceptions of the value and originality of fashion items. Through a variety of art-related activities, such as working with artists, recruiting artistic directors, receiving sponsorships, fundraising, and marketing, the connection between fashion and the arts is developed.

Artification ([Massi and Turrini 2020](#); [Crane 2019](#); [Kastner 2014](#); [Kapferer 2014](#); [Mendes and Rees-Roberts 2015](#)) refers to the process of turning something that was once considered utilitarian or functional into something that is now considered art or art-like. It involves breaking the traditional boundaries of fashion, and incorporating elements of creativity, imagination, and cultural significance into the design process.

---

[4] "The essence of the luxury brands is the identity, which is how the customers perceive the brands and who the brands are in reality", see ([Nässel and Persson 2011](#); [Ricca and Robins 2012](#)).

The concept of artification of fashion has gained momentum in recent years as more designers and fashion houses have focused on creating pieces that are not only functional, but also aesthetically pleasing and meaningful.

The artification of fashion can take many different forms, such as the use of high-quality materials, intricate designs, unique shapes or patterns, and the incorporation of artistic elements into clothing or accessories.

For instance, the French luxury brand Chanel is known for its iconic tweed jackets (Chanel 2023), which have been elevated to works of art with intricate embroidery, beading, and other embellishments.

It can also involve collaborations between fashion designers and artists, where the two fields come together to create pieces that blur the lines between fashion and art (Black and Veloutsou 2017; Kapferer 2014; Pullig et al. 2006; Zorloni 2016; Krim 2022; Baumgarth 2018; Chailan 2018; Codignola and Rancati 2016; Dion and Arnould 2011; Jelinek 2018; Kapferer and Bastien 2012).

Overall, the artification of fashion reflects a growing appreciation for the artistic value of clothing and accessories, as well as a desire to elevate fashion beyond its practical functions and into the realm of art.

Here are a few examples of collaborations between artists and haute couture brands:

1. Louis Vuitton x Jeff Koons: in 2017, Louis Vuitton collaborated with artist Jeff Koons to create a collection of handbags and accessories featuring Koons' iconic artwork, including his "Gazing Ball" series. The bags featured Koons' reproductions of famous paintings, such as the Mona Lisa and Van Gogh's "Wheat Field with Cypresses."

2. Dior x Marc Quinn: in 2019, Dior collaborated with British artist Marc Quinn to create a collection of haute couture dresses featuring his artwork. Quinn's designs featured colourful, nature-inspired prints, including his "Garden" and "In the Night Garden" series.

3. Gucci x Unskilled Worker: Italian fashion house Gucci collaborated with artist Unskilled Worker (real name: Helen Downie) to create a collection of clothing and accessories featuring her colourful, whimsical artwork. The collection included Gucci's signature handbags and sneakers, adorned with Unskilled Worker's playful designs.

4. Alexander McQueen x Sarah Lucas: in 2008, British fashion designer Alexander McQueen collaborated with artist Sarah Lucas to create a collection of avant-garde dresses featuring Lucas' provocative, sexually-charged artwork. The dresses were made from unconventional materials, such as stockings and lace, and featured graphic prints and bold silhouettes.

Exhibiting fashion items in museums and galleries is one tactic in an anticommodification plan. This strategy has guaranteed fashion brands the protection of copyright law, as demonstrated by the example of Italian legislation. The other option is to use appropriated works of art in one's own fashion designs.

As to designers, they are focused on both trends and customer expectations. Customers set the standard for high-quality, sustainable, materials-based goods, as well as inspiration from the arts. Nevertheless, it is significant that brands must follow established trends. They construct items with the intention to market them. The idea of "art for art's sake" has no foundation here. Vera Wang asserts that she is "never very commercial in her ready-to-wear lines" since her distinctive style combines elements of both art and fashion (Beard 2019). Miuccia Prada highlighted the conflict between commerce and art by saying: "[I]deas can be so pure when you do the fashion show, but my job forces me to see the bad things—'This doesn't work; this isn't selling'. It forces you to see the reality, and to understand what people like, even when that isn't always what you like yourself. That is the most relevant point in my work: always to face reality. When it is good that is fine—it doesn't make my life better—but I only care about what doesn't work" (Wingfield 2016).

## 4. Divergent Thinking in Copyright Law

Divergent thinking is becoming increasingly important in the fashion industry as consumers demand more unique and innovative products. Brands and designers who can think outside the box and create products that challenge traditional fashion are more likely to succeed in the future.

Similarly, adapted divergent thinking appears in art. Divergent thinking is an important skill for artists to develop, as it may help them overcome creative blocks, stay inspired, and create innovative and engaging work. The ability to generate a wide range of ideas and perspectives, whether working alone or in collaboration with others, can be an asset in the art world. It seems that twentieth-century artists (Marcel Duchamp, Fernand Léger, and others) have known about it for a long time. When incorporating the Mona Lisa in their works, they got into discussions with their predecessor, respecting the existence of the original.

For Marcel Duchamp, it became a pretext for questioning the concept of the work of art. The piece is part of Duchamp's "readymade" series, which entailed taking existing objects and modifying them in some way to create a new work of art. L.H.O.O.Q. exemplifies Duchamp's use of irony and humour in his art (The Guardian 2001), as well as his interest in challenging conventional notions of art and beauty. Duchamp's addition of the moustache and goatee transformed the iconic image of the Mona Lisa into a humorous and irreverent commentary on the nature of art and the artist's role.

Similarly, when Fernand Léger painted the Mona Lisa with keys (Sassoon 2001), he turned the Mona Lisa into a machine-like figure, with her face replaced by a series of mechanical gears and her hair and clothing simplified into geometric shapes. The keys in the painting allude to the idea of discovering the secrets hidden behind the Mona Lisa's enigmatic smile. He was implying a kind of equivalence: having identified modernity with a democracy of visual forms (New York Times 1998), he meant to say that the Mona Lisa and the keys were equally valuable, equally familiar, and equally modern. Leger challenged the notion of the "perfect" human form, implying that the future of art lies in the embrace of new technologies and modern aesthetics.

As can be observed, often, rather than a complete break with the past, artists enter a polemic with their predecessors while creating something new. "Access to creativity is therefore a condition for new creativity" (Cohen 2006; Morrison 2006; Guibault and Hugenholtz 2006).

However, a key concept for the placement of artworks in fashion is the "public domain". The accepted common understanding of the *domaine public* means that an object in it does not so much belong to anyone as belong to everyone. Works "falling into the public domain" (fr. *tomber dans le domaine public* (Choisy 2002)) that are deprived of legal protection due to an expired protection period become widely available and, more importantly, it is possible to use them without restriction.

Most designers take inspiration from artists' works in a broad sense, not only paintings, sculptures, or utilitarian objects, but also decorative interior elements. Both the decorative element, i.e., the artistic, original glass mirror and Vincent van Gogh's painting "Iris", inspired Yves Saint Laurent in the creation of some of the most outstanding collections in his entire artistic *oeuvre*, the haute couture S/S 1988 (Musée Yves Saint Laurent 2023a) and 1990 (Musée Yves Saint Laurent 2023b) collections.

The Iris evening jacket in organza, entirely embroidered with sequins, enriched with beads, pearls, tubular pearls, painting, and painted ribbons, is the French designer's homage to the eminent painter, Vincent van Gogh. Although the Dutch artist's work has long since entered the public domain, it has not only been given new life, but has become a new work of art thanks to the painstaking work of the *brodeur*.[5]

---

[5]   (en. *embroiderer; a person who embroiders*); embroidery is the skilled technique of embellishing and decorating a garment by hand, using stitches in silks and yarns and sometimes including sequins, beads, feathers and pearls. Professional embroiderers are masters of detail, applying a range of traditional stitching techniques to produce intricate designs on clothing, accessories, and home décor items; see more: (Mauriès 2020; Pale 2018; Albertini 2021).

Van Gogh's famous "Sunflowers" was also reinterpreted in the same collection. The free use of works from the public domain provokes and serves to make designers even more intellectual, as well as fulfilling its original purpose—creativity becomes free and universally accessible.

Furthermore, artists also collaborated with each other and often used divergent thinking in their work. For example, François-Xavier Lalanne and his wife Claude Lalanne were French sculptors who frequently incorporated elements of nature and animals into their work. François-Xavier Lalanne created a mirror for Yves Saint Laurent's Paris apartment (Sotheby's 2019), which one day inspired the French designer to create a unique jacket.

"One day Yves Saint Laurent called out to François Lesage 'come and see'. He rushed over. Showing him the reflection of the crystal chandelier and the Parisian sky in a mirror by Lalanne, Saint Laurent said 'this is what I want'. Monsieur Lesage returned with three versions that captured morning, midday, and evening light. 'Wonderful', said Yves Saint Laurent crying, 'we will do all three. 350 h of embroidery each'" (Benaïm 2018).

In addition to the decorative elements, one can see, among other things, the exploitation of Lalanne's work and its inclusion in the design of the jacket; however, it should be noted that the mirror is not in the public domain—the heirs of the artists who created the mirror for Yves Saint Laurent are still entitled to copyright protection (Sotheby's 2023).

## 5. Iconoclasm *en vogue*. The Louis Vuitton and Christian Louboutin Paradox

While economic copyrights are harmonised at the EU level, moral copyrights, unfortunately, are not. In the European Union, moral rights are a set of rights granted to authors or creators of original works that are distinct from the economic rights associated with copyright. First, the right of paternity: the author has the right to be identified as the creator of the work and to object to any distortion, mutilation, or modification of the work that would harm their reputation.

Second, the right of integrity: the author has the right to object to any distortion, mutilation, or modification of the work that would harm its integrity. These rights are not transferable, and last for the author's lifetime and even after death. The author's moral rights are considered inalienable in the EU, which means they cannot be waived or sold. Injunctions, damages, and public apologies may be issued for violations of moral rights.

In this context, the Court of Justice of the European Union, in the Eva-Maria Painer[6] case, held that Article 5(3)(d) of Directive 2001/29/EC establishes an obligation to indicate, when quoting, the source together with the name. Although the described case relates to the right of quotation, it should be added that this obligation is also incumbent on the exploiter of the work, even if the piece is in the public domain. This principle is seen in the works of the *maison* Louis Vuitton, which emblazoned on its classic handbag models such as the Speedy and Neverfull reproductions of the works of the great painting masters of the ages such as da Vinci, Monet, Manet, van Gogh, Rubens, Titian, or Gauguin, and prominently printed the artists' names alongside (Vogue British 2017).

It could be argued that Louis Vuitton, as a French brand (with France as the cradle of copyright), should pay particular attention to respecting its own and others' copyrights. However, the latest reports seem to contradict this. "The Joan Mitchell Foundation sent a cease-and-desist letter to the Paris headquarters of Louis Vuitton on Tuesday, alleging the fashion brand had used the artist's paintings in handbag advertisements after her nonprofit organisation repeatedly declined to give its approval" (New York Times 2023).

As the Joan Mitchell Foundation points out, "in accordance with its longstanding policy that images of the artist's work be used only for educational purposes" (France24 2023), and it is "a grave disappointment that Louis Vuitton has such disregard for the rights of an artist and would exploit her work for financial gain" (*Idem*). Joan Mitchell's heirs are demanding that the advertising campaign be withdrawn, or they will take legal action,

---

6　Judgment of the Court of Justice of the European Union (Third Chamber) of 1 December 2011, in case C–145/10, *Eva-Maria Painer*, ECLI:EU:C:2013:138.

according to a statement from the Foundation. Contacted by Agence France Presse, Louis Vuitton representatives did not comment. On social networks, as well as on its website, the advertising campaign no longer appears (Le Figaro 2023). Can we therefore speak of blatant iconoclasm[7] and copyright infringement? It seems to be, but it remains to be seen how things will develop.

Art in the service of an advertising campaign also appeared at Christian Louboutin. Collaborations with photographer Peter Lippmann resulted in campaigns inspired by the surrealist work of Rene Magritte (Vogue France 2013) or portrait paintings (Glamour UK 2011). This prompts the question of legal assessments of such actions. While there can be no question of an infringement of copyright because most works are in the public domain, the commercial purpose, and the advertised object (shoes) recall the right to integrity of the work, as well as the right to fair use.

It can be concluded that the lack of harmonisation of rules at the EU level (Brown 2020) may cause legal issues, such as whether the incorporation of a work of art constitutes an infringement or is merely inspiration. An interplay of two works, primary and secondary, of authorship can come in many shades, and it is never easy for lawyers to assess whether consent is required from the primary copyright holder. The possible categories are: inspiration, a work with borrowings, derivative work, plagiarism. The lingering question is also whether 'fair use' (or 'permissible use' under the civil law concept) can constitute a legitimate excuse for appropriation art.

## 6. Dante's Nine Circles of Hell: A Few Words about Art Appropriation

Appropriation art (Schaumann 2015; Mix 2015; McLeod and Kuenzli 2011) is a type of contemporary art that involves taking pre-existing images, objects, or ideas from popular culture, art history, or everyday life, and incorporating them into a new artwork with a different context, meaning, or purpose. Appropriation artists often use various techniques such as copying, reproducing, recontextualising, or combining different elements to create a new work that challenges the traditional notions of originality, authorship, and value.

Appropriation art emerged in the 1960s and 1970s as a response to the increasing influence of mass media, consumer culture, and globalisation on contemporary art. It was also a way for artists to critique the dominant cultural, social, and political ideologies of their time by subverting, parodying, or deconstructing the mainstream representations and narratives. Some well-known examples of appropriation art include Andy Warhol's Brillo Boxes, Marcel Duchamp's Fountain, and Sherrie Levine's After Walker Evans.

Appropriation art (Tate Modern 2023), which began to be widely used by artists in the 1980s, contributed not only to the popularisation of high art, but also opened a veritable Pandora's box with its ambivalent attitude towards copyright. The art of appropriation is not, by its proponents, referred to as plagiarism, but only as recontextualization (Oxford Reference 2023), i.e., the placing of other people's works in a different, new context (Linden 2016), particularly, in a copyright law.

The relationship between appropriation art and copyright law can be complex and contentious (Hamilton 2021). Copyright law grants the creator of an original work exclusive rights to control the use and distribution of their work, which includes the right to prevent others from reproducing, adapting, or publicly displaying the work without permission.

Appropriation art often involves using pre-existing works that are protected by copyright, which can raise legal issues of infringement and fair use. In some cases, appropriation artists may argue that their use of copyrighted works is transformative, meaning that it creates a new work with a different meaning or purpose that is not simply a copy of the original. Transformative use may be considered fair use under copyright law, which allows

---

7   *Iconoclasm* literally means "image breaking" and refers to a recurring historical impulse to break or destroy images for religious or political reasons. For example, in ancient Egypt, the carved visages of some pharaohs were obliterated by their successors; during the French Revolution, images of kings were defaced. In the context of the Joan Mitchell vs. Louis Vuitton case, it may be considered iconoclastic to use images commercially without the consent of the copyright owner; see more: (Gamboni 2007; Paic 2021; Françon 1968).

limited use of copyrighted material without permission for purposes such as criticism, commentary, or parody.

Overall, the relationship between appropriation art and copyright law highlights the tension between creative expression and intellectual property rights, and raises important questions about the nature of artistic innovation, cultural appropriation, and the limits of copyright protection in the digital age.

Not all appropriation art may be considered transformative, and determining the legal boundaries of fair use can be subjective and context-dependent. As a result, appropriation artists may face legal challenges or accusations of plagiarism or copyright infringement, and the legality of their works may be disputed in courts or through settlement agreements. Examples of copyright-related issues raised in the past are:

1.  "Campbell's Soup Cans" by Andy Warhol (Sotheby's 2018)—This artwork consists of a series of paintings featuring images of Campbell's Soup cans. Warhol's use of the Campbell's Soup label was considered an act of appropriation, as he used a pre-existing commercial image as the basis for his artwork. The Campbell Soup Company did not initially approve of Warhol's use of their trademark and threatened legal action, but ultimately did not pursue a lawsuit.
2.  "The Last Supper" by Rene Magritte (The Menil Collection 2023)—In this artwork, Magritte appropriates Leonardo da Vinci's famous painting "The Last Supper" by replacing the figures with blank white sheets. This act of appropriation raised questions about the limits of copyright protection for public domain artworks.
3.  "Untitled (Cowboy)" by Richard Prince (MET 2023a)—This artwork consists of a photograph of a cowboy taken from a Marlboro cigarette advertisement, which Prince rephotographed and enlarged. The photographer who took the original Marlboro photo sued Prince for copyright infringement, but the case was settled out of court.
4.  "After Walker Evans" by Sherrie Levine (MET 2023b)—In this artwork, Levine reproduces photographs taken by the American photographer Walker Evans and presents them as her own work. Levine's appropriation of Evans' photographs raised questions about originality and authorship in contemporary art.

One of the most recognisable creators of appropriation art is Jeff Koons. Not only has the artist been the subject of numerous lawsuits, but he has also caused resentment, particularly among the French. The resentment is primarily related to numerous copyright infringements of other artists in the context of the exploitation of their artworks. In a 2017 case, the French *Tribunal de grande instance de Paris* found that "the work Naked, a 1988 porcelain sculpture by Jeff Koons, is a forgery of a photograph of two naked children taken in 1970 by Jean-François Bauret".[8]

The Paris court of consequence ruled Jeff Koons and the Centre Georges Pompidou guilty and ordered them to pay EUR 40,000 to the family of the deceased photographer, half of which was used to cover court costs. The defendants had to pay an additional EUR 4000 for the use of the photograph of the sculpture on their website. An important point is that the defendants did not dispute that the sculpture was inspired by a photograph. Jeff Koons, however, tried to give priority to freedom of artistic expression under Article 10 ECHR (Council of Europe 1950). According to the same thesis, the Centre Pompidou wanted to promote freedom of public information.

It should be added that an artist such as Jeff Koons is a walking paradox: on the one hand, a notorious plagiarist, while on the other, his works are valued at auction at exorbitant sums—his work Balloon Dog (Orange) was sold for nearly USD 58.4 million (Christie's 2013).

A small breakthrough was the judgment of the Court of Cassation in the Klasen[9] case, in which a significant decision was made, as it relied on the freedom of expression guaranteed by Article 10 ECHR, and considered that the creative freedom of the author of a derivative work should be considered. Judges who rule on the interplay between creative

---

8　TGI Paris, 9 mars 2017, n° 15/01086.
9　Cass. civ. 1ère, 5 mai 2015, «Klasen», n° 13-27391.

freedom and copyright monopoly should find the right balance, the concrete balance between the protection of the copyright of the original work and the artistic freedom of the author of the derivative work, without, however, undermining the possibility for judges to find infringement by the author of the derivative work if necessary.

Nevertheless, the exceptions contained in Article L-122-5 of the CPI,[10] i.e., short quotation or parody, were rejected in the present case. The judges rejected the right of brief quotation because the photographs covered too large an area of the images in which they were incorporated (between 20% and 56% of the second work), as well as the parody exception for a lack of ridiculing or comic elements.

In addition, reference was made to the case law of the Court of Justice, which, in determining the concept of parody, ruled that "it must be interpreted that the essential characteristics of parody are, firstly, that it evokes an existing work while being noticeably different from it and, secondly, that it constitutes an expression of humour or mockery".[11] It should therefore be emphasised that, where an unauthorised use of a protected work does not fall within one of these exceptions, the dependent work is in principle infringing.

Even outside the European continent, infamous plagiarism of works by street-art artists also happens, even at major fashion houses. Roberto Cavalli was sued by a group of graffiti artists (Revok, Reyes, and Steel) from Northern California for using their works in his Graffiti collection without their permission (Vogue British 2014). In their lawsuit, the artists not only demanded damages, but pointed to the moral harm they had suffered due to the association of European chic and glamour with the street artist. The Italian designer, refuting these allegations, claimed that he had not copied someone else's work, but had been inspired by it. "According to court papers, Revok, Reyes, and Steel filed to voluntarily dismiss their cases, although specific details were not disclosed" (Vogue British 2016).

Appropriation art, which, due to its inherent appropriation of original artworks in its own work, is receiving wide criticism, also in the fashion industry. Appropriation art has been used in fashion in a variety of ways, often as a means of commenting on or subverting prevailing cultural and commercial trends. Here are some different types of appropriation in fashion:

1. **Upcycling** (Aus et al. 2021): upcycling involves taking existing materials, garments, or other objects and transforming them into new, higher-value products. This can involve repurposing vintage or secondhand clothing, using leftover materials from manufacturing processes, or creatively reimagining outdated or otherwise discarded items.
2. **Logomania** (Cochrane 2018): logomania refers to the trend of prominently featuring logos or brand names on clothing and accessories, often to the point of excess. This trend has been appropriated and subverted by artists and designers who use logos and branding in unexpected or ironic ways, or who create their own logos to comment on consumer culture.
3. **Collage** (Vaughan 2005): collage is a common technique in appropriation art and can be used in fashion to combine disparate elements into a new and unexpected whole. This can involve cutting and pasting images, fabrics, and other materials together, or using digital tools to create collages that blur the boundaries between traditional media.
4. **Deconstruction** (Kanters 2018): deconstruction involves taking apart and reassembling existing garments or materials in unconventional ways. This can result in designs that challenge traditional ideas about fit, form, and function, and that create unexpected silhouettes and shapes.
5. **Subversive embroidery** (Parker 2010): embroidery has been used in fashion for centuries, but some contemporary designers have used it in subversive ways, incorporating political messages, irreverent humour, or unexpected imagery into their designs.

---

[10] Le code de la propriété intellectuelle est un document du droit français, créé par la loi n°92-597 du 1er juillet 1992 relative au code de la propriété intellectuelle, publié au Journal officiel du 3 juillet 1992.

[11] Judgment of the Court of Justice of the European Union of 3 September 2014, in case C-201/13 *Deckmyn*, ECLI:EU:C:2014:2132.

6. **Subversive branding** (Kuanr et al. 2022): subversive branding involves taking elements of traditional branding and using them in unexpected or subversive ways. This can involve creating new logos or slogans that challenge consumer culture, or using familiar branding elements in new and unexpected ways.

7. **Remixing** (Lascity 2019): remixing involves taking existing designs or motifs and reinterpreting them in new and different ways. This can involve combining different styles or eras, incorporating unexpected materials or techniques, or subverting traditional notions of gender, race, or class.

8. **Remixing cultural references** (Knobel and Lankshea 2008): fashion designers often draw inspiration from different cultures and historical periods, but some designers take this a step further by combining cultural references in unexpected ways, creating designs that challenge traditional notions of authenticity and cultural appropriation.

9. **Sampling** (Sloboda et al. 2001): sampling involves taking elements from different sources and remixing them to create something new. In fashion, this can involve taking fabrics, prints, or motifs from different cultures or time periods and combining them in new and unexpected ways.

10. **Photocollage** (Diakopoulos and Essa 2005): photocollage is a technique that involves combining photographs or photographic elements into a single image. In fashion, this can involve taking images of different garments, fabrics, or accessories and combining them in new and unexpected ways to create a new design.

11. **Print mixing** (Golobic et al. 2019): print mixing is the practice of combining different patterns and prints in a single garment or outfit. This can involve mixing prints from different cultures, time periods, or design traditions, and can result in designs that challenge traditional ideas about colour, shape, and texture.

12. **Found object fashion** (Zborowska 2017): found object fashion involves using nontraditional materials, such as recycled or repurposed objects, in fashion design. This can involve using materials such as plastic bottles, old clothing, or even trash to create new and innovative designs that challenge traditional notions of luxury and materiality.

These are just a few examples of the ways in which appropriation art has been used in fashion, often as a means of challenging prevailing norms and expectations, and creating new forms of expression. They often challenge traditional notions of style, culture, and identity.

In conclusion, appropriation in art and fashion can be a source of conflict, especially when artists feel that their work is being exploited or devalued. However, there are also opportunities for collaboration and creativity when the two fields intersect, respectfully and mutually beneficially, to be used as a means of expression.

### 7. Botticelli's Venus, a Bone of Contention between Jean Paul Gautier and Uffizi Gallery

Venus, painted in 1480 by Botticelli, is renowned as an Italian Renaissance masterpiece of immeasurable creative significance. The French fashion house Jean Paul Gaultier was reportedly being sued by the Uffizi Museum in October 2022 for using a depiction of Botticelli's Venus on apparel without the museum's consent (Abogados 2023). Evidently, a warning to that effect had been disregarded. On dresses, blouses, trousers, and scarves in the Gaultier collection, Venus' likeness may be seen. The company also used pieces from other artists' works, including those of Rubens and Michelangelo. The Uffizi Gallery is suing the French fashion brand for EUR 100,000 in damages for using a piece of art that belongs to it. The Uffizi bases its legal action on the requirement that "the use of images of Italian public property must be expressly authorised and a fee must be paid" as stated in Italy's 2004 law on the protection of monuments. As noted by Monereo Meyer Abogados, the works in question are already in the "public domain", therefore, even though Italy has such specific cultural heritage legislation that condemns Gaultier's behaviour, from a copyright perspective it is legally feasible to make use of it.

## 8. Legal Discussion

The question that remains relates to the specificity of the Italian law's basis, and whether it is possible to sue for the use of a "public domain" work of art without the owner's permission. Periodically, museums' claims in that regard make legal news. As noted by Wojciech Kowalski, while embracing the right to enjoy heritage in principle, it should be kept in mind that, like every right, it has some restrictions because, if exercised by some people, it may potentially violate the rights of others (Kowalski 2021, 2022). The Faro Convention acknowledges this issue and emphasises in this regard that "exercise of the right to cultural heritage may be subject only to those restrictions which are necessary in a democratic society for the protection of the public interest and the rights and freedoms of others". He determines that the legal restrictions can be related to one or both of the following aspects of the legal status of the work: proprietary rights and copyrights. As the latter can be excluded in this matter, the remaining legal basis is the legal concept of property. This dates back to Ancient Rome, when Roman lawyers gave this construct three tenets: (1) right to possess (lat. *ius possidendi*), (2) right to use, right to enjoy, and right to fruits (lat. *ius utendi, fruendi et abutendi*), and (3) right to dispose (lat. *ius disponendi*). The last two entitlements in particular can be gauged as the basis of a legal action. As Wojciech Kowalski concluded:

> "it should be emphasized upfront that this is of key importance for the present discussion, as it also includes the right to dispose of the appearance of monuments, which is closely related to their photographic recording. Incidentally, it should also be emphasized that this refers only to the appearance of things that are directly and visually perceptible, which should by no means be equated with the concept of "image." This concept under Polish Law equates to "personal image" and as such belongs to the category of moral rights and is reserved exclusively for natural persons" (Kowalski 2022).

This raises the question of whether "personal image" and "property rights" also refer to appearances that have been preserved in a photograph, a motion picture or another medium. Given the absolute clarity of the Italian law, it is difficult to refute such claims by owners. If, as previously stated, there is no barrier to property owners monetising their asset in this way, then the income derived through the direct use of an object's appearance should likewise be regarded as benefits derived from that object, even if it was recorded on a medium separate from the object.

Appropriation can potentially be considered fair use under certain circumstances, but it depends on the specific context and nature of the use. Interestingly, on a European basis, it is worth noting Polish law and its unique construction of permitted use. The fair use exception in Polish copyright law is an important aspect of promoting cultural expression and creativity because it allows for the use of copyrighted material in certain contexts without the copyright owner's permission (Barta et al. 2011).

This is especially true in the realm of culture, where works of art, literature, music, and other forms of creative expression are frequently influenced and built upon by previous works. Journalists and critics, for example, may use excerpts from copyrighted works in their reporting and analysis, and educators may use copyrighted material in their lessons and lectures. The fair use exception also helps to preserve and promote cultural heritage. Archives and libraries can ensure that important cultural works are accessible and available for future generations by allowing the use of copyrighted material for preservation purposes.

Overall, the fair use exception in Polish copyright law is an important tool for encouraging cultural expression and creativity, while still protecting the rights of copyright holders. As indicated in the literature, fair use is an expression of the legislator's realisation that private use of works is inevitable and uncontrolled, whereas public use is an expression of the state's appropriate educational and cultural policy (Barta et al. 2011).

Legal regulations should consider an individual's access to previous cultural output in each field of creativity (Błeszyński 1985). It permits the use of copyrighted material in certain circumstances while still protecting the economic and moral rights of creators and copyright holders. Despite the fact that works accessible in the open space under Article 33(1) of the Copyright Act[12] are immaterial creations of the human mind that find the material dimension in fixed form, such as buildings or monuments, they serve as testimony to culture in the open space. However, because there is no legal definition of "culture", it may be understood in practice either as cultural heritage or as the concept of cultural property as a source of identity or to national cultural heritage (Chałubińska et al. 2018). The fair use exception allows individuals and organisations to engage with and comment on works of culture in meaningful ways by allowing the use of copyrighted material for purposes such as review, criticism, education, and reporting current events.

In the United States, the fair use doctrine allows limited use[13] of copyrighted material without permission for purposes such as criticism, commentary, news reporting, teaching, scholarship, or research. In order to determine whether a particular use qualifies as fair use, four factors are considered:

1. The purpose and character of the use—transformative uses that create a new work with a different meaning or purpose are more likely to be considered fair use than uses that merely copy or replicate the original work.

2. The nature of the copyrighted work—the more creative or original the work, the less likely it is to be considered fair use.

3. The amount and substantiality of the portion used—using only a small or insignificant portion of the original work is more likely to be considered fair use than using the entire work or a significant portion of it.

4. The effect of the use on the potential market or value of the original work—uses that have little or no impact on the market for the original work are more likely to be considered fair use than uses that directly compete with or diminish the market for the original work.

In the case of appropriation art, some courts have found that certain uses of copyrighted material are transformative and therefore qualify as fair use, while others have not. Ultimately, whether a particular use of copyrighted material is considered fair use depends on the specific context and facts of the case.

Here are a few examples of court cases in the United States where appropriation art has been at the centre of fair use disputes:

1. Blanch v. Koons (2006)[14]—in this case, photographer Andrea Blanch sued artist Jeff Koons for copyright infringement after Koons used one of her photographs as the basis for a sculpture. Koons argued that his use of the photograph was a transformative fair use, but the court disagreed and found that the sculpture did not sufficiently transform the original photograph.

2. Cariou v. Prince (2013)[15]—in this case, photographer Patrick Cariou sued artist Richard Prince for copyright infringement after Prince used several of Cariou's photographs in a series of paintings. Prince argued that his use of the photographs was transformative, but the court initially ruled in favour of Cariou. The decision was later partially reversed on appeal, with the court finding that some of Prince's works were transformative and therefore constituted fair use.

3. Graham v. Prince (2018)[16]—in this case, artist Donald Graham sued Richard Prince for copyright infringement after Prince used one of Graham's Instagram posts as the basis for a series of paintings. Prince argued that his use of the post was transformative and,

---

[12] Article 29 of PCL; Law on Copyright and Related Rights of 4 February 1994 (consolidated text J.L. of 2017, item 880, as amended), hereinafter: PCL.
[13] Fair Use Doctrine, 17 U.S.C.S. § 107 (1977).
[14] Blanch v. Koons, 467 F.3d 244, 246 (2d Cir. 2006).
[15] Cariou v. Prince, 714 F.3d 694 (2d Cir. 2013).
[16] Graham v. Prince—265 F. Supp. 3d 366 (S.D.N.Y. 2017).

therefore, a fair use, but the court found that the paintings did not sufficiently transform the original post.

These cases show how courts have struggled to apply the fair use doctrine to appropriation art, with some works being deemed transformative and, therefore, fair use, while others are not. Ultimately, the determination of whether a particular work constitutes fair use depends on the specific context and nature of the use.

There have been several court cases in the United States related to appropriation art in the fashion industry. Here are a few examples:

1. Leiber v. Warner Bros. Entertainment Inc. (2008)[17]—fashion designer Judith Leiber sued Warner Bros. for copyright infringement after the company used images of her handbags in the movie "The Devil Wears Prada." Warner Bros. argued that its use of the images was fair use, but the court found that the use was not transformative and therefore did not qualify as fair use.

2. Rentmeester v. Nike, Inc. (2015)[18]—photographer Jacobus Rentmeester sued Nike for copyright infringement after the company used one of his photographs of Michael Jordan in a Jumpman logo. Nike argued that its use of the photograph was transformative and, therefore, a fair use, but the court found that the use was not sufficiently transformative and therefore did not qualify as fair use.

3. Louis Vuitton Malletier, S.A. v. Warner Bros. Entertainment Inc. (2012)[19]—luxury fashion brand Louis Vuitton sued Warner Bros. for trademark infringement after the company used a knockoff version of a Louis Vuitton handbag in the movie "The Hangover Part II." Warner Bros. argued that its use of the bag was fair use, but the court found that the use was not sufficiently transformative and therefore did not qualify as fair use.

4. Morris v. Young (2010)[20]—artist Dan Eldon's family sued retailer Anthropologie and artist Samantha Margaret Young for copyright infringement after they used images from Eldon's journals on clothing and other merchandise. Young argued that her use of the images was transformative and, therefore, a fair use, but the court found that the use was not transformative enough and therefore did not qualify as fair use.

These cases demonstrate how courts have applied the fair use doctrine to appropriation art in the fashion industry, with some works being found to be transformative and, therefore, a fair use, while others are not.

### 9. Conclusions

It goes without saying that France is the cradle not only of copyright law, but also of fashion law. It would therefore seem proper to believe that French brands should be the ones to set an example for legal compliance, that is, capitalising on and monetising the creativity of others in a fair way. This research proves the opposite. The recent *Joan Mitchell Foundation* vs. *Louis Vuitton* case blatantly contradicts this assumption and proves that the most prominent national brands disregard legal constraints when it comes to the fashion business and branding. The top-notch fashion brands, with their phenomenal revenues and impressive big-business legal teams, play the game of the jungle, where the strong entity does not submit to the weak one, regardless of the moral (legal) code.

Legally speaking, the heterogeneity of regulation at the EU level of important aspects of moral rights, permitted use, or rights management can lead to manifold infringements. Divergent thinking may cause various legal issues in cases of inspiration by previous works—among many, the major vexing question is what sort of interaction is there between the primary and secondary works. The variety of relevant legal qualifications—inspiration, work with borrowings, derived work, or plagiarism—make the answer truly hard. Legal

---

17  Warner Bros. Entm't Inc. v. RDR Books—575 F. Supp. 2d 513 (S.D.N.Y. 2008).

18  Rentmeester v. Nike, Inc. —883 F.3d 1111 (9th Cir. 2018).

19  Louis Vuitton Malletier, S.A. v. Warner Bros. Ent., 1-11-CV-09436-ALC-HBP (S.D.N.Y.).

20  Morris v. Young—925 F. Supp. 2d 1078 (C.D. Cal. 2013).

subsumption is not always easy, leaving appropriation art in a very uncertain legal state. Additionally, it is hard to argue that appropriation art constitutes a creative activity within the meaning of copyright law.

Overall, while copyright law can sometimes present challenges to preserving and promoting cultural heritage, there are also opportunities for designers, brands, and cultural institutions to work together to create innovative and sustainable practices that benefit both creators and the public, thanks to divergent thinking.

In the fashion industry, branding plays a crucial role in establishing a brand's identity and reputation, and is often protected by copyright law. Copyright law protects original works of authorship, including visual elements such as logos, graphics, and other design elements used in branding. This safeguard ensures that others do not misappropriate or misrepresent a brand's identity and reputation.

Furthermore, copyright law can protect specific aspects of a brand's design and product features, such as a product's overall design or the unique elements that distinguish it from other similar products. This can help to prevent competitors from copying or imitating a work, which is especially important in the fashion industry, where trends and designs are frequently copied. It should be noted, however, that copyright protection is not absolute, and has limitations. Copyright protection, for example, does not extend to functional elements of a product, such as the shape or design of a garment required to achieve a specific function. Furthermore, copyright protection may be limited by the principle of fair use or other exceptions to copyright law that allow for certain uses of copyrighted material without permission.

In general, branding in the fashion industry is an important aspect of developing and maintaining a successful brand identity, and is frequently protected by copyright law. However, designers and brands must be aware of the copyright law's limitations and exceptions.

This paper also refers to works of authorship in the public domain, that, despite their legal status, also trigger legal questions as to their use and monetisation. It is generally argued in the legal literature that, next to copyright, property rights are the other legal concept that offers a monopoly to the owner, no matter whether the owner is a private person or a museum.

Overall, copyright, cultural heritage, and branding are inextricably linked and play important roles in constructing our cultural landscape. The literature emphasises that "the digital revolution has dramatically increased the ability of individuals and corporations to appropriate and profit from the cultural knowledge of indigenous peoples" (Brown 1998). It is critical to achieve a balance between safeguarding creators' rights, preserving cultural heritage, and encouraging branding innovation and creativity.

In conclusion, it should be highlighted that, in contrast to other sectors of the creative economy, copying in fashion is strikingly pervasive, enforcement against fakes is oddly lenient, and the line between original work and imitation is extraordinarily fine, so much so that it can be challenging to tell the difference between several possible situations: genuine inspiration, a popular grassroots trend, and copying.

Consequently, observers of fashion law witness a perfect melting pot of authors' copyrights violations, both moral and economic.

**Author Contributions:** Conceptualization, M.J. and B.S.; methodology, M.J.; software, M.J.; validation, M.J. and B.S.; formal analysis, M.J.; investigation, M.J.; resources, M.J. and B.S.; data curation, M.J. and B.S.; writing—original draft preparation, M.J. and B.S.; writing—review and editing, M.J. and B.S.; visualization, M.J. and B.S.; supervision, M.J.; project administration, M.J.; funding acquisition, M.J. All authors have read and agreed to the published version of the manuscript.

**Funding:** This research was funded in whole by National Science Centre in Poland, grant "Brand Abuse: Brand as a New Personal Interest under the Polish Civil Code against an EU and US Backdrop", grant holder: Marlena Maria Jankowska-Augustyn, number:2021/43/B/HS5/01156. For the purpose of Open Access, the author has applied a CC-BY public copyright licence to any Author Accepted Manuscript (AAM) version arising from this submission.

**Institutional Review Board Statement:** Not applicable.

**Informed Consent Statement:** Not applicable.

**Data Availability Statement:** We used generally accessible research data.

**Conflicts of Interest:** The authors declare no conflict of interest. The funders had no role in the design of the study; in the collection, analyses, or interpretation of data; in the writing of the manuscript; or in the decision to publish the results.

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
