# Peer review of "From Fashion Brand to Artwork: Divergent Thinking, Copyright Law, and Branding"

_laws, 2023_

Round 1

Reviewer 1 Report

I fail to see a connection between many of the section titles and the content thereof. The interaction between copyright, branding, marketing and heritage protection is also unclear.

Author Response

Dear Reviewer, 

thank you for your comments. The relation between copyright, branding and marketing has been made more clear now. The manuscript underwent a substantial change will be uploaded today. 

Best regards,

Authors 

Reviewer 2 Report

Generally, the paper is well structured, research methods and objectives are well set. However with regards to the objective of the paper, I would advise the author to indicate the geographical focus (limitation) of the paper. Furthermore, I would like to bring to the attention of the author that in some cases footnotes are not complete or there are typos (…); see footnote n. 12, n. 52.

Author Response

Dear Reviewer, 

thank you for your comments. The relation between copyright, branding and marketing has been made more clear now. The footnotes were added and corrected.

The manuscript underwent a substantial change will be uploaded today. 

Best regards,

Authors 

Reviewer 3 Report

The paper discusses brands, copyright and cultural heritage in the fashion field.

It is unclear how the notion of convergence is reflected all along the paper. The work refers to brands, but it never mentions trademarks, which are a crucial tool when discussing IP and thus next to copyright.

There is confusion between Italian provisions on cultural heritage (not monuments) and copyright at lines 80-84.

There is no consistency when discussing copyright: as an example lines 166 169: if there is no copyright, there is no need to discuss of fair use. And also, it is better to avoid jumping from economic rights to moral rights and back without explaining the reason of that reference

The explaination for public domain is inaccurate. The connections between public domain, copyright and cultural heritage provisions are not clearly expressed. The impact that these rules and their combination have on the images reproduced on fashion item is not described in a structured and sound fashion.

Overall, the paper needs to be restructured. It needs to clearly present the guiding line since its introduction and to specify which legal framework it will study. The author has to make sure that relevant literature in law is used and that any time that a legal provision is referred, it is clearly explained and it is used for supporting the defended position.

Author Response

Dear Reviewer,

Thank you for your comments. The relation between copyright, branding, and marketing has become clearer now. However, due to space limitations, trademark law was excluded, as mentioned in the paper. Also, 'convergence' as a concept has been removed.

As explained in the paper, the "fair use" or "permitted use" doctrine is crucial when considering the exception of fashion law vs. artistic expression copyright.

The manuscript underwent substantial changes and will be uploaded today.

Best regards,

Authors 

Round 2

Reviewer 3 Report

The revised version of the paper contain some clearer descriptions than the former one. However, the effort made is still not sufficient for the publication.

It is not possible to put branding, cultural heritage and copyright at the same level (and say that they have IPRs implication). Branding is not a legal term, thus the connection should be established with TM strategy. Cultural heritage is not a form of protection, while copyright is.

Cultural heritage as defined by the paper is not often protected by IP. On the contrary copyright cannot apply for two reasons at least. There is no consistency between the first notion of cultural heritage and the one that seems to be suggested in a copyright perspective

Overall, it is still unclear what is the thesis of the author exactly and how the different notions and arguments serve it

The EU copyright framework is not so heterogeneous, it is harmonized as to the economic rights. It is true that exceptions and limitations are not harmonised across countries.

When the author uses the term "Directive": which directive(s) are we talking about?

The statements made on moral rights are inexact (not sure if it is a matter of wording used or a conceptual mistake).

Author Response

Hello,

please find our response attached. 

Warm regards,

Marlena
